# Through-the-Needle Biopsy Revisited: How Patient Selection and Standardization Reduce Adverse Events in Pancreatic Cyst Evaluation

**DOI:** 10.3390/diagnostics15243096

**Published:** 2025-12-05

**Authors:** Maria Cristina Conti Bellocchi, Maria Vittoria Teso, Sofia Spagnolo, Erminia Manfrin, Sokol Sina, Antonio Pea, Nicolò de Pretis, Roberto Salvia, Luca Frulloni, Stefano Francesco Crinò

**Affiliations:** 1 Diagnostic and Interventional Endoscopy of Pancreas, Pancreas Institute, University of Verona, 37134 Verona, Italy; stefanofrancesco.crino@aovr.veneto.it; 2 Gastroenterology Unit, Department of Medicine University of Verona, 37134 Verona, Italynicolo.depretis@univr.it (N.d.P.); luca.frulloni@univr.it (L.F.); 3Department of Diagnostics and Public Health, Section of Pathology, University of Verona, 37134 Verona, Italysokol.sina@aovr.veneto.it (S.S.); 4 Unit of Pancreatic Surgery, Department of Engineering for Innovation Medicine, Pancreas Institute, University of Verona, 37134 Verona, Italy; antonio.pea@univr.veneto.it (A.P.); roberto.salvia@univr.it (R.S.)

**Keywords:** IPMN, endoscopic ultrasound, fine-needle aspiration, mucinous cysts, pancreatic cancer

## Abstract

**Background/Objectives**: Pancreatic cystic lesions (PCLs) are increasingly being detected due to the widespread use of cross-sectional imaging. Endoscopic ultrasound (EUS) is the preferred modality for evaluating their nature and malignancy risk, yet fluid analysis and cytology offer limited sensitivity. Through-the-needle biopsy (TTNB) has emerged as a more accurate diagnostic tool, though it is associated with higher adverse event (AE) rates. In 2021, our center implemented a selective TTNB protocol excluding frail or elderly patients and suspected IPMNs and standardizing the procedure to two passes, complete cyst aspiration, and selective antibiotic prophylaxis. This study aimed to compare AE rates before and after protocol implementation, evaluate safety factors including antibiotic use, and assess TTNB adequacy and diagnostic accuracy. **Methods**: We retrospectively analyzed consecutive patients referred for TTNB at AOUI Verona between March 2016 and March 2025, dividing them into two groups: before (Group A) and after (Group B) protocol adoption. Patients not punctured due to technical issues, lack of indication, or presumed pseudocystic nature were excluded. **Results**: Of 970 patients evaluated by EUS, 190 underwent TTNB (100 in Group A and 90 in Group B). Lesions were mainly located in the pancreatic body or tail, with a significantly larger size in Group B. The overall AE rate was 6.3%, significantly higher in Group A (11%) than in Group B (1%). Antibiotic prophylaxis was not associated with AE occurrence. TTNB adequacy was 88.9%, and diagnostic accuracy was 75.3%. Among 68 surgical cases, TTNB was accurate in 79.4%. **Conclusions**: A selective and standardized TTNB approach significantly reduces AEs while maintaining high adequacy and diagnostic accuracy.

## 1. Introduction

Pancreatic cystic lesions (PCLs) are increasingly identified in clinical practice, necessitating accurate diagnostic assessments [1]. Endoscopic ultrasound (EUS) is regarded as the most precise diagnostic procedure for evaluating PCLs, offering a detailed description of morphology and serving as a guide for fine-needle aspiration (EUS-FNA) [2]. However, the diagnostic performance of EUS-FNA in distinguishing between mucinous and non-mucinous cysts, and more importantly, identifying the histotype of the PCL along with its associated risk of malignancy, is often unsatisfactory [3,4]. To address these limitations, a novel microforceps device (Moray™, US Endoscopy, Ohio, USA) has been introduced, enabling through-the-needle biopsy (TTNB) of cyst walls via a standard 19G EUS-FNA needle. TTNB has demonstrated superior diagnostic sensitivity and accuracy (75–99.2%) compared to EUS-FNA, though it carries a notable risk of adverse events (AEs), including severe complications [5,6,7,8,9,10]. A multicenter retrospective study of 506 patients undergoing TTNB reported an overall AE rate of 11.5%, identifying three risk categories: high risk (IPMN, multiple passes; 28% AE rate), low risk (patients < 64 years, non-IPMN, ≤2 passes, complete aspiration; 1.4%), and intermediate risk (remaining cases; 6.1%) [11]. These findings prompted a shift in our clinical practice toward a selective TTNB protocol, emphasizing patient stratification and procedural standardization. This study evaluates the impact of this tailored approach on AE rates and diagnostic performance, aiming to optimize the risk–benefit balance and support more effective clinical management of PCLs.

## 2. Materials and Methods

### 2.1. Patient Selection

Consecutive patients who underwent EUS for pancreatic cystic lesions (PCLs) at the Endoscopic Unit of the Pancreas Institute of Verona between March 2016 and March 2025 were evaluated. The cohort was divided into two groups based on the timing of the TTNB procedure relative to protocol implementation in January 2021. Group A included patients who underwent TTNB before March 2021, under non-standardized conditions. Group B included patients who underwent TTNB after January 2021, following a selective and standardized protocol.

The standardized protocol introduced the following procedural modifications: TTNB was not performed in patients with suspected IPMN or in frail/elderly individuals; the number of microforceps passes was limited to two; complete cyst aspiration was performed whenever feasible; antibiotic prophylaxis was selectively administered based on clinical judgment. Frailty was defined according to operator judgment, typically corresponding to patients with numerous comorbidities or severe anesthesiological risk. Elderly patients were generally considered those >80 years, especially when comorbidities were present. Patients not punctured due to technical limitations, lack of indication, or presumed pseudocystic nature were excluded. Patients who were excluded from TTNB but underwent fine-needle biopsy (FNB) or EUS-FNA were not included in the present analysis, as their evaluation falls outside the scope of this study.

### 2.2. Aims

The primary endpoint was the rate of adverse events (AEs) following TTNB, defined according to the international lexicon and classified using AGREE criteria [12].

The secondary outcomes were categorized as follows:Calculated across the entire TTNB population:-Cytohistological adequacy: Defined as the proportion of TTNB samples sufficient for histological interpretation, per American Gastroenterology Association white paper [13].-Diagnostic accuracy: Defined as the concordance between TTNB diagnosis and final diagnosis (based on surgical pathology in patients who underwent surgery or composite clinical/imaging/histological criteria).Comparison between Group A and Group B:-Diagnostic yield: Defined as the proportion of EUS procedures that provided a clinically actionable diagnosis (e.g., mucinous vs. non-mucinous cyst), per American Gastroenterology Association white paper [13].-Impact on clinical management: Assessed as the proportion of patients in whom TTNB findings influenced decision-making (e.g., surgical indication, cessation of surveillance, definitive diagnosis in unilocular cysts).Exploratory analysis:-Utility of antibiotic prophylaxis: Evaluated by comparing AE rates in patients who received antibiotics versus those who did not [14].

### 2.3. Procedures

TTNB procedures were performed by expert endosonographers as described below. After puncturing the cyst with a 19 G EUS-FNA needle, the stylet was removed, and a small amount of fluid was aspirated for biochemical analysis. The microforceps were then introduced through the needle, and biopsies were performed on the cyst wall. Retrieved specimens were extracted from the microforceps jaws using the metal hook provided with the forceps, transferred between two 16 mm diameter paper disks (code 100623, Milestone srl, Sorisole, BG, Italy), and finally introduced into a cassette. Each specimen was placed separately in formalin vials. Samples were fixed in 10% formaldehyde solution and embedded in paraffin, sectioned at 5 μm, and stained with hematoxylin and eosin (H&E). Supplementary slides were prepared for histochemical and immunohistochemical staining. At the end of the TTNB procedure, the cystic fluid was, whenever possible, completely aspirated. Part of the fluid was used for analysis and dosage of markers (amylase, glucose, CEA), part of the fluid was centrifuged, and the sediment was smeared onto slides and processed according to Papanicolaou staining.

### 2.4. Statistical Analysis

Descriptive statistics were used. Continuous variables were presented as mean (±standard deviation [SD]) or median (with interquartile range [IQR] and ranges) and were compared using the Wilcoxon rank-sum (Mann–Whitney) test if the assumptions of the t-test were not respected (i.e., if data were not normally distributed). Categorical variables were expressed as frequencies (%) and compared using the χ^2^ or the Fisher exact test when appropriate. Univariate and multivariate analyses were performed to evaluate factors impacting the safety of TTNB.

To assess the potential impact of operator experience on TTNB safety, we analyzed the temporal distribution of adverse events throughout the study period. This allowed us to evaluate whether AE rates decreased over time, independent of protocol changes, and to contextualize the learning curve associated with TTNB adoption.

## 3. Results

### 3.1. Patient Population

Between March 2016 and March 2025, 970 patients were referred for EUS evaluation of pancreatic cystic lesions (PCLs). After applying exclusion criteria, 190 patients who underwent TTNB were included: 100 in Group A (pre-protocol) and 90 in Group B (post-protocol). The remaining patients either underwent FNA/FNB or were not punctured; their evaluation falls outside the scope of this study. The flowchart is summarized in Figure 1.

The mean age of TTNB patients was 52.3 ± 14.8 years, with a predominance of female patients (74%). The primary indication for EUS was lesion size >3 cm with unclear etiology (78.4%), followed by suspected mural nodules or thickened walls (21.6%). Baseline characteristics were comparable between groups (Table 1), except for a higher proportion of suspected mucinous cysts on cross-sectional imaging in Group B (*p* = 0.004).

### 3.2. Procedural Details

Most lesions were located in the pancreatic body or tail (68%), with a mean size of 46.7 ± 19.5 mm. Unilocular morphology and thin walls were predominant. TTNB was performed using a standardized technique, with a median of three passes in Group A and two in Group B. Complete cyst aspiration was achieved in 69.5% of cases. However, in six patients, three in group A and three in group B, only one pass was performed due to a technical issue in one, an anesthesiological reason in one, the choice of endoscopist in two, and intraprocedural bleeding in two. Antibiotic prophylaxis was more frequently administered in Group A (87%) than in Group B (68.9%, *p* = 0.002). Detailed procedural data are shown in Table 2.

### 3.3. Fluid Analysis and Cytology

Fluid analysis was available in 140 patients (73.7%). CEA levels were diagnostic (>192 or <5 ng/mL) in 72 patients (51.4%), while glucose and amylase levels were informative in 60% and 57.3% of cases, respectively. However, only 35.8% of patients had concordant markers sufficient to classify cyst type. Cytology was performed in 165 patients, yielding a diagnostic result in 40% of cases.

### 3.4. Primary Outcome

The overall AE rate was 6.3%, significantly higher in Group A (11%) than in Group B (1%, *p* = 0.005). A post hoc power analysis confirmed that the study was adequately powered (>87%) to detect this difference. Adverse events included pancreatitis, infection/fever, and bleeding requiring hospitalization. To assess the impact of operator experience, AE rates were analyzed over time, revealing a clustering of events between 2018 and 2021, prior to protocol standardization. Moreover, 25 (13.1%) patients, 15 (15%) in group A and 10 (11.1%) in group B, experienced intraprocedural self-limiting and uneventful bleeding that were managed conservatively and were classified as “incidents” according to the AGREE classification. A detailed list of AEs is reported in Table 3.

Univariate and multivariate analyses were conducted to identify predictors of AEs. Group A allocation and final diagnosis of mucinous cysts were independently associated with increased AE risk (odds ratio [OR] 11.5, 95% confidence interval [CI] 1.40–94.07; *p* = 0.023 and *p* = 0.025, respectively). Antibiotic use and complete aspiration were not significant predictors (Table 4).

### 3.5. Secondary Outcomes

TTNB samples were adequate for histological interpretation in 88.9% of cases, with no significant difference between groups (Group A: 92%, Group B: 86.7%; *p* = 0.904). Among twenty patients with inadequate TTNB results, nine patients remained undefined (seven because they were lost on follow-up and two with stable features on surveillance), and four intraductal papillary mucinous neoplasms (IPMNs), two serous cystic neoplasms, two pseudocysts, one mucinous cystic neoplasm, one simple mucinous cyst, and one solid pseudopapillary neoplasm were eventually diagnosed, six of them after surgery. Diagnostic accuracy, assessed in 68 surgical cases, was 79.4% overall. TTNB outperformed cytology (42.6%) and fluid markers (54.4%) in diagnostic concordance. Excluding 17 patients with undefined lesions, TTNB demonstrated a sensitivity of 82.6%, specificity of 83.3%, and overall accuracy of 82.6%. Although Group B showed numerically higher performance, differences were not statistically significant (Table 5).

### 3.6. Outcomes and Follow-Up

After TTNB, 68 (35.8%) patients underwent surgery: 44 (44%) in group A and 24 (26.6%) in group B (*p* = 0.01). TTNB was accurate in 54 (79.4%) patients, biomarkers in 37 (54.4%), and cytology in 29 (42.6%). Details of inaccurate TTNB cases (*n* = 14, 20.6%), nine in group A and five in group B, are reported in Table A1. In non-resected patients, 20 (16.3%) patients were lost to follow-up, and 102 underwent surveillance, with a mean follow-up of 31.9 ± 28.3 months. A final diagnosis of IPMN was eventually made in sixteen patients, eight in group A and eight in group B. All final diagnoses, with TTNB accuracy and surgical outcome, are reported in Table 6.

## 4. Discussion

The widespread use of cross-sectional imaging has led to a rising detection of PCLs, intensifying the need for accurate risk stratification and histological characterization. While EUS-FNA remains a cornerstone in initial evaluation, its limited diagnostic yield—particularly in differentiating mucinous from non-mucinous cysts—has prompted the adoption of TTNB as a more reliable alternative [15,16,17]. However, TTNB carries a non-negligible risk of AEs, estimated at 7% in recent meta-analyses [5], especially in patients with suspected IPMN or when multiple passes are performed.

In light of prior evidence showing increased AE rates in IPMN and with extensive sampling [11], our center progressively adopted a selective TTNB protocol. This approach includes pre-procedural stratification based on age, frailty, and imaging suspicion. Procedural steps were also standardized: microforceps passes were limited to two, complete cyst aspiration was performed when feasible, and antibiotic prophylaxis was reserved for selected cases. When IPMN was suspected, alternative tissue acquisition techniques such as FNB or cytology on aspirated fluid were preferred.

Our findings confirm that this tailored strategy significantly improves safety without compromising diagnostic performance. The AE rate dropped from 11% in Group A to 1% in Group B (*p* = 0.005), with a post hoc power analysis (>87%) supporting the robustness of this result. This difference remained significant even when including intraprocedural bleeding events, most of which were mild and self-limiting. Only one bleeding episode required prolonged hospitalization and was classified as a true AE. Other complications included pancreatitis and cyst infection with fever. Interestingly, neither antibiotic prophylaxis nor complete aspiration alone explained the reduction in AEs. This highlights the importance of combining patient selection with procedural standardization. Our results align with the propensity score-matched study by Facciorusso et al. [18], which found no significant benefit of antibiotic prophylaxis in preventing infectious complications after TTNB. The high rate of complete cyst aspiration in our series may further explain the lack of association.

To evaluate whether operator experience influenced AE rates, we analyzed their temporal distribution. No increase in complications was observed following team expansion, suggesting that protocol standardization had a greater impact on safety than individual experience. Multivariate analysis identified Group A allocation and mucinous cyst diagnosis as independent predictors of AEs. Viscous cystic content and intraprocedural bleeding may impair pancreatic fluid outflow, especially in lesions that communicate with the ductal system. These factors can also hinder complete aspiration. Importantly, reducing the number of TTNB passes in Group B did not compromise sample adequacy (89%) or diagnostic accuracy (82%).

Our data support performing TTNB with two passes and complete aspiration in unilocular or oligocystic lesions. Antibiotic prophylaxis may be reserved for cases with incomplete aspiration or bleeding [14]. When combined with fluid analysis, TTNB yielded actionable diagnostic information in most cases. In resected patients, concordance with the final diagnosis was high. In contrast, TTNB should be avoided in suspected ductal communication or multifocal IPMN, where FNA or FNB may offer safer alternatives. Overall, tissue acquisition should be reserved for cases where it is expected to influence clinical management. In patients with a clear surgical indication, additional sampling may be unnecessary and potentially harmful, exposing patients to procedural risks and theoretical concerns about seeding [19].

This study has several limitations that should be acknowledged. First, its retrospective and single-center design may limit the generalizability of the findings, as patient selection and procedural expertise could differ across institutions. Second, the definition of frailty relied on operator judgment, typically corresponding to patients with multiple comorbidities or severe anesthesiological risk, which introduces a degree of subjectivity. Third, the relatively low number of AEs resulted in wide confidence intervals for some odds ratio estimates, indicating limited precision. Finally, pathologists were not formally blinded to TTNB diagnoses during histopathologic correlation, which may have introduced bias. Despite these limitations, the large cohort size and systematic protocol implementation strengthen the validity of the observed reduction in AEs and support the reproducibility of the selective TTNB approach.

## 5. Conclusions

In conclusion, our study demonstrates that selective patient inclusion and procedural standardization in TTNB for PCLs significantly enhance safety while maintaining high diagnostic adequacy and accuracy. FNA and FNB remain valuable tools in selected scenarios, particularly in frail patients or when IPMN is suspected. This approach offers a reproducible framework for optimizing diagnostic yield while minimizing risk.

## Figures and Tables

**Figure 1 diagnostics-15-03096-f001:**
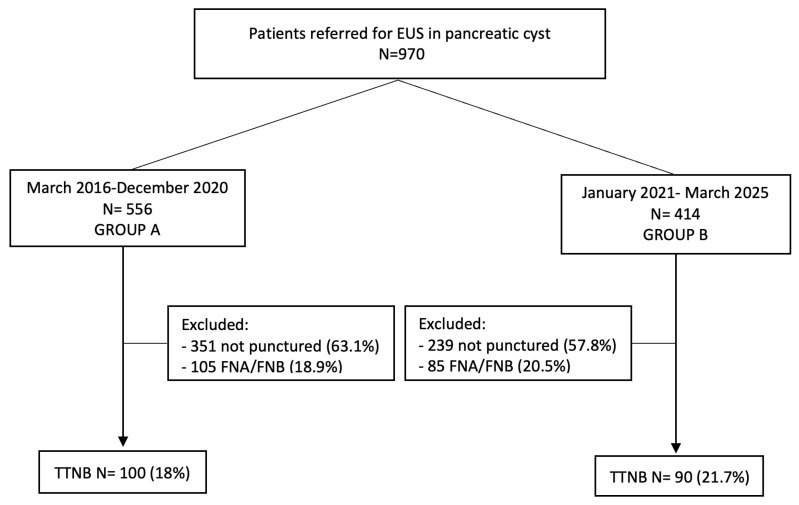
Study flowchart. EUS, endoscopic ultrasound; FNA, fine-needle aspiration; FNB, fine-needle biopsy; TTNB, through-the-needle biopsy.

**Table 1 diagnostics-15-03096-t001:** Baseline characteristics of 190 patients included in the study, comparing Group A (under non-standardized conditions) and Group B (selective and standardized protocol).

Variables	Overall *n* = 190	Group A *n* = 100	Group B *n* = 90	*p* Value
Age (years), mean ± SD	52.1 ± 14.7	51.2 ± 14.9	53.5 ± 14.8	0.288
Sex				0.208
Male	49 (26%)	22 (22%)	27 (30%)	
Female	141 (74%)	78 (78%)	63 (70%)	
BMI (kg/m^2^)	25.3 ± 4.9	25.5 ± 5	25.1 ± 4.9	0.790
CCI	1	1	1	NS
Imaging performed				NS
US	88 (47.1%)	59 (60.9%)	29 (32.2%)	
CT	102 (54.5%)	43 (44.3%)	59 (65.6%)	
MRI	162 (86.6%)	88 (90.7%)	74 (82.2%)	
Previous EUS	28 (15%)	11 (11.3%)	17 (18.9%)	
Size on cross-sectional (mm)				
Mean ± SD	48.2 ± 3.5	45.7 ± 17.6	51.2 ± 28	0.114
Suggested diagnosis on cross-sectional imaging *				
				0.004
Serous cystic neoplasm	48 (25.2%)	31 (31%)	17 (18.8%)	
Mucinous cyst	93 (48.9%)	59 (59%)	34 (37.7%)	
Not specified	68 (35.2%)	28 (28%)	40 (44.4%)	
Other diagnoses	6 (3.1%)	1 (1%)	5 (2.6%)	

* more than one diagnosis could be suspected on cross-sectional imaging. *SD*, standard deviation; *BMI*, Body Mass Index; *CCI*, Charlson Comorbidity Index; *NS*, not significant; *US*, ultrasound; *CT*, Computed Tomography; *MRI*, Magnetic Resonance Imaging; *EUS*, endoscopic ultrasound.

**Table 2 diagnostics-15-03096-t002:** Procedural details of endoscopic ultrasound-guided through-the-needle biopsy.

Variables	Overall *n* = 190	Group A *n* = 100	Group B *n* = 90	*p* Value
Size of lesion (mm) on EUS				
Mean ± SD	46.7 ± 19.5	44 ± 16.6	49.8 ± 22	0.044
Site of lesion				0.349
Head/uncinate process	56 (29.5%)	32 (32%)	24 (26.7%)	
Body/tail	127 (66.8%)	66 (66%)	61 (67.7%)	
Extra pancreatic	7 (3.7%)	2 (2%)	5 (5.6%)	
Morphology				0.061
Unilocular	109 (57.4%)	51 (51%)	58 (64.4%)	
Oligocystic	81 (42.6%)	49 (49%)	32 (35.6%)	
Cyst walls				0.263
Thin	130 (68.4%)	72 (72%)	58 (64.4%)	
Thickened	60 (31.6%)	28 (28%)	32 (35.6%)	
Cyst content				0.118
Anechoic	100 (52.6%)	58 (58%)	42 (46.7%)	
Inhomogeneous	90 (47.4%)	42 (42%)	48 (53.3%)	
Intracystic lesion				
No	167 (87.9%)	83 (83%)	84 (93.3%)	
Yes	23 (12.1%)	17 (17%)	6 (6.7%)	
Thickened septum/nodule	6/17	6/11	0/6	0.029
TTNB passes				
≤2 passes	74 (38.9%)	25 (25%)	49 (54.4%)	<0.001
≥3 passes	116 (61.1%)	75 (75%)	41 (45.6%)	
Visible specimens (median)	2			
Complete cyst aspiration				
Yes	132 (69.5%)	69 (69%)	63 (70%)	
No	58 (30.5%)	31 (31%)	27 (30%)	0.881
Antibiotic administration				
Yes	149 (78.4%)	87 (87%)	62 (68.9%)	
No	41 (21.6%)	13 (13%)	28 (31.1%)	0.002

*SD*, standard deviation; *EUS*, endoscopic ultrasound.

**Table 3 diagnostics-15-03096-t003:** Adverse events occurring in the study cohort.

Patient, Sex (Years)	Group A/B	Adverse Event	Severity *	Final Diagnosis
Female (57)	A	Fever/infection	II	MCN
Female (27)	A	Hypotension	I	MCN
Female (40)	A	Mild acute pancreatitis	II	MCN
Female (47)	A	Hematoma	I	SCN
Male (75)	A	Fever/infection	II	Undefined
Female (70)	A	Fever/infection	II	Undefined
Female (62)	A	Orticaria	II	SCN
Male (75)	A	Severe acute pancreatitis (ICU admission)	IIIb	IPMN
Female (63)	A	Necrotic pancreatitis with infected collections	IV	MCN
Female (65)	A	Xantogranulomatous reaction/gastric mass	IIIb	MCN
Female (45)	A	Mild pancreatitis	II	MCN
Female (65)	B	Mild pancreatitis	II	IPMN

* according to the AGREE classification [12]. *MCN*, mucinous cystic neoplasm; *SCN*, serous cystic neoplasm; *ICU*, intensive care unit; *IPMN*, intraductal papillary mucinous neoplasm.

**Table 4 diagnostics-15-03096-t004:** Univariate and multivariate analysis of factors associated with adverse events.

Variables	Univariate Analysis		Multivariate Analysis	
	AE Yes/No	*p* Value	*p* Value	OR (95% CI)
Group		0.004	0.023	11.5 (1.4–94.1)
A	11/89			
B	1/89			
Complete aspiration		0.833	-	
Yes	5/127			
No	3/55			
Antibiotic administration		0.646	-	
Yes	6/143			
No	2/39			
Final diagnosis		0.062	0.025	6.2 (1.2–32.1)
Mucinous	8/75			
Non-mucinous	2/96			
Unknown	2/17			

*AE*, adverse event; *OR*, odds ratio; *CI*, confidence interval.

**Table 5 diagnostics-15-03096-t005:** Diagnostic performance of endoscopic ultrasound-guided through-the-needle biopsy.

	Overall	Group A	Group B	*p* Value
Sensitivity, % (95% CI)	82.6% (76–88)	79% (68.9–87.1)	86.4% (77–93.2)	0.152
Specificity, % (95% CI)	83.3% (35.8–99.5)	66.7% (9.4–99.1)	100% (15.8–100)	0.438
Accuracy, % (95% CI)	82.6% (76.2–87.9)	78.6% [68.7–86.6]	86.7% (77.5–3.2)	0.124

*CI*, confidence interval.

**Table 6 diagnostics-15-03096-t006:** Final diagnosis and outcome of patients included in the study.

Final Diagnosis	Number (%)	Accuracy of TTNB	Surgery
SCN	60 (31.6%)	54/60 (90%)	1/60 (1.6%)
MCN	58 (30.5%)	53/58 (91.4%)	44/58 (75.8%)
IPMN	16 (8.6%)	5 /16 (31.2%)	4/16 (25%)
SPN	2 (1%)	1/2 (50%)	2/2 (100%)
Rare histotype	31 (16.3%)	26/31 (83.9%)	14/28 (50%)
Pseudocyst	6 (3.2%)	4/6 (66.7%)	3/6 (50%)
Undefined	17 (8.9%)	-	-

*TTNB*, Through-the-needle biopsy; *SCN*, serous cystic neoplasm; *MCN*, mucinous cystic neoplasm; *IPMN*, intraductal papillary mucinous neoplasm; *SPN*, solid pseudopapillary mucinous neoplasm.

## Data Availability

The data presented in this study are available on request from the corresponding author. The data are not publicly available due to privacy restrictions.

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
