# Peer review of "Through-the-Needle Biopsy Revisited: How Patient Selection and Standardization Reduce Adverse Events in Pancreatic Cyst Evaluation"

_diagnostics, 2025, doi:10.3390/diagnostics15243096_

Round 1

Reviewer 1 Report

Comments and Suggestions for Authors

I would like to congratulate the authors on this well-written manuscript. The article is well presented and methodologically robust, from the Verona Pancreas Institute,  evaluating how a standardised, selective TTNB protocol impacts safety and diagnostic performance in pancreatic cystic lesions (PCLs). The study compares outcomes before and after introducing this protocol in 2021, incorporating specific procedural refinements (two passes, complete cyst aspiration, and selective antibiotic use).
The authors report a substantial reduction in adverse events (AEs) from 11% to 1%, while maintaining high diagnostic adequacy (88.9%) and accuracy (82.6%).

This is an important contribution to the field, particularly given increasing reliance on TTNB in complex PCL management. The work is clinically relevant, clearly presented, and addresses a key gap in the literature on standardisation and safety optimisation of TTNB. However, I have the following comments for the authors to address.

  1. The authors present a large and meticulously analysed cohort; however, it is a single-centre retrospective series, which should be explicitly highlighted as a limitation. It would be valuable to provide more details on the selection criteria used to define “frail” or “elderly” patients excluded from the standardised protocol. Was this based on Charlson Comorbidity Index, performance status, or operator judgment?
  2. The use of the AGREE classification is acceptable, but the authors need to specify whether all AEs were adjudicated prospectively or retrospectively from procedure notes. Can the authors also clarify whether asymptomatic biochemical pancreatitis (elevated amylase without clinical symptoms) was included or excluded.
  3. The reduction in AE rate (11% → 1%) is somewhat striking. The authors should expand on how operator experience was controlled for in this study since procedural standardisation and increasing expertise might both contribute.
  4. The finding that antibiotic use did not significantly affect AE rates contrasts with some prior meta-analyses. Please discuss possible explanations (did high cyst aspiration rate mitigate infection risk?).
  5. For the 68 surgical cases, was the pathologist blinded to the TTNB diagnosis during histopathologic correlation?
  6. The discussion would benefit from a paragraph contextualising how this selective approach could be implemented in practice
  7. Some odds ratios (OR 11.5; CI 1.4–94.1) have very wide confidence intervals. Please mention in the discussion that these estimates may be imprecise due to low AE event counts.
  8. There are few minor typographical mistakes that need addressing. For example, “thickened” misspelled in one table; “hystotype” should be “histotype”.

Author Response

  1. The authors present a large and meticulously analysed cohort; however, it is a single-centre retrospective series, which should be explicitly highlighted as a limitation. It would be valuable to provide more details on the selection criteria used to define “frail” or “elderly” patients excluded from the standardised protocol. Was this based on the Charlson Comorbidity Index, performance status, or operator judgment?

RE: Thank you for your suggestions. We agree that the single‑centre and retrospective design represent a limitation of our study. This point has now been explicitly acknowledged in a dedicated Discussion section.

RE: We thank the reviewer for this important comment. In our study, the definition of “frail” patients was based primarily on operator judgment, which in practice corresponded to individuals with multiple comorbidities or with a severe anesthesiological risk profile. The CCI is not routinely recorded and the retrospective design did not allow a meticulous evaluation of this data.  Elderly patients were generally considered those >80 years, particularly when associated with relevant comorbidities. This clarification has been added to the Methods section.

The use of the AGREE classification is acceptable, but the authors need to specify whether all AEs were adjudicated prospectively or retrospectively from procedure notes. Can the authors also clarify whether asymptomatic biochemical pancreatitis (elevated amylase without clinical symptoms) was included or excluded.

RE: We thank the reviewer. All adverse events were adjudicated retrospectively from procedure notes and electronic records, according to AGREE criteria. Asymptomatic biochemical pancreatitis (isolated amylase elevation without clinical symptoms) was excluded, in line with international definitions.

  1. The reduction in AE rate (11% → 1%) is somewhat striking. The authors should expand on how operator experience was controlled for in this study since procedural standardisation and increasing expertise might both contribute.

RE: We agree. To address this, we analyzed the temporal distribution of AEs across the study period. The clustering of events before 2021 suggests that protocol standardization, rather than operator experience alone, accounted for the reduction.

  1. The finding that antibiotic use did not significantly affect AE rates contrasts with some prior meta-analyses. Please discuss possible explanations (did high cyst aspiration rate mitigate infection risk?).

RE: We thank the reviewer for this insightful comment. In our cohort, antibiotic prophylaxis was not significantly associated with reduced infectious adverse events. A possible explanation is the high rate of complete cyst aspiration, which may have mitigated. infection risk regardless of antibiotic use. Our findings are consistent with the propensity score‑matched study by Facciorusso et al. (Diagnostics 2022; 12:211), which demonstrated that antibiotic prophylaxis does not significantly influence infection rates after TTNB of pancreatic cysts. We have added this reference and expanded the discussion accordingly.

  1. For the 68 surgical cases, was the pathologist blinded to the TTNB diagnosis during histopathologic correlation?

No, pathologists were not blinded. We added this fact as a potential limitation.

  1. The discussion would benefit from a paragraph contextualising how this selective approach could be implemented in practice

RE: We have added a paragraph in the Discussion outlining how selective TTNB can be integrated into clinical practice, emphasizing patient stratification, IPMN exclusion, limiting passes, and reserving TTNB for cases where results are expected to change management.

  1. Some odds ratios (OR 11.5; CI 1.4–94.1) have very wide confidence intervals. Please mention in the discussion that these estimates may be imprecise due to low AE event counts.

RE: We agree. The wide CIs reflect the low number of AE events and should be interpreted with caution. This limitation has been noted in the Discussion.

  1. There are few minor typographical mistakes that need addressing. For example, “thickened” misspelled in one table; “hystotype” should be “histotype”.

RE: We thank the reviewer for noting these errors. They have been corrected in the revised manuscript.

Reviewer 2 Report

Comments and Suggestions for Authors

The Title and Abstract summarize the main aspect of the work.

The Introduction contains data regarding the importance of endoscopic ultrasound (EUS) in the diagnosis of pancreatic cystic lesions, together with fine needle aspiration (FNA) and through-the-needle biopsy (TTNB) of the cyst walls characterisation. The higher accuracy of TTNB, but also the higher risk of adverse events, compared to the FNA, is also emphasized. For this reason, a better standardization of the protocol for TTNB can allow a significant reduction of adverse events without impairing the accuracy of pathological results.

The Methods described by this study are clear and replicable. The timeframe of the study was adequate. The selected group of patients included 190 patients with EUS and TTNB performed, divided into 100 cases before standardization and 90 cases after standardization. The statistical methods were accurately described. No concerns related to the study were noted.

The Results are clearly and transparently presented. The percentage of adverse events, defined by the AGREE classification, was significantly higher in patients performing TTNB before standardization, whereas the sensitivity, specificity, and accuracy regarding the pathological diagnosis were similar. in multivariate analysis, only the group A allocation (before standardization was introduced) and mucinous cyst lesion were independent predictors for the presence of adverse events, which supported the standardization of EUS-TTNB as an important measure in order to reduce the rate of adverse events in patients with pancreatic cystic lesions.

Minor observations: in Line 156, there is a missing “t” from the significant word. Table 2 is wrongly numbered as Table 1. In the last row of Table 4, an OR is missing.

Important observation: in the Results section, there were 11 patients with adverse events in group A and one in group B, whereas in Table 3 (which details characteristics of patients with adverse events), 12 patients in group A and one in group B were recorded.

In the Discussions section of the paper, the authors analyzed the importance of a standardized EUS-TTNB for the reduction of adverse events. The findings obtained by the authors support the introduction of a tailored strategy for safety improvement without impairing the diagnostic performance rate of EUS-TTNB. The importance of TTNB with two passes during the EUS examination is emphasized by the authors and supported by multivariate analysis findings.

The Conclusions are clear.

References are adequate for the purpose of the paper and are correctly cited.

Tables contain clear and legible information. Several observations regarding Tables 2, 3, and 4 were mentioned above in my observations regarding the Results section.

No concerns about the validity of the study were detected.

Author Response

Minor observations: in Line 156, there is a missing “t” from the significant word. Table 2 is wrongly numbered as Table 1. In the last row of Table 4, an OR is missing.

Important observation: in the Results section, there were 11 patients with adverse events in group A and one in group B, whereas in Table 3 (which details characteristics of patients with adverse events), 12 patients in group A and one in group B were recorded.

We sincerely thank the reviewer for carefully pointing out several minor issues. All typographical errors, table misnumbering (Table 2 corrected), and the missing odds ratio in Table 4 (now reported as OR 6.2, 95% CI 1.2–32.1) have been addressed. In addition, the discrepancy between the number of adverse events in Group A reported in the Results section (11 patients) and Table 3 (12 patients) has been corrected, with the accurate figure being 11. The manuscript has been thoroughly revised to ensure consistency and accuracy across text and tables
